# A GENERALIZED PROBABILITY KERNEL ON DISCRETE DISTRIBUTIONS AND ITS APPLICATION IN TWO-SAMPLE TEST

## ABSTRACT

We propose a generalized probability kernel(GPK) on discrete distributions with finite support. This probability kernel, defined as kernel between distributions instead of samples, generalizes the existing discrepancy statistics such as maximum mean discrepancy(MMD) as well as probability product kernels, and extends to more general cases. For both existing and newly proposed statistics, we estimate them through empirical frequency and illustrate the strategy to analyze the resulting bias and convergence bounds. We further propose power-MMD, a natural extension of MMD in the framework of GPK, illustrating its usage for the task of two-sample test. Our work connects the fields of discrete distribution-property estimation and kernel-based hypothesis test, which might shed light on more new possibilities.

## 1 INTRODUCTION

We focus on the two-sample problem, which is given two i.i.d samples $\{x_1, x_2, ...x_n\}$, $\{y_1, y_2, ..., y_n\}$, could we infer the discrepancy between underlying distributions they are drawn from. For such a problem, the option of hypothesis test(two-sample test) is most popular, and a variety of statistics in estimating the discrepancy is proposed. In recent years, RKHS based method such as maximum mean discrepancy(MMD) has gained a lot of attention. (Gretton et al., 2012) has shown that in a universal-RKHS $\mathcal{F}$, $\mathrm{MMD}(\mathcal{F}, \boldsymbol{p}, \boldsymbol{q}) = 0$ if and only if $\boldsymbol{p} = \boldsymbol{q}$, thus could be used for the two-sample hypothesis test. (Gretton et al., 2012) further provides unbiased estimator of MMD with fast asymptotic convergence rate, illustrating its advantages.

On the other hand, estimating distribution properties with plugin(empirical) estimators on discrete setting is an active research area in recent years, where people focus on problem settings with large support size but not so large sample size. The Bernstein polynomial technique is introduced to analyze the bias of the plugin estimators in (Yi & Alon, 2020), which provides remarkable progress on bias-reduction methods of the plugin estimators. It is thus interesting to ask if the plugin estimators could motivate new results for the RKHS-based two-sample test.

Another interesting topic is about the probability kernel, defined as kernel function over probabilities, instead of over samples. As is easily seen, any discrepancy measure of distribution $\boldsymbol{p}$ and $\boldsymbol{q}$ could potentially be valid probability kernels, not so much work focuses on this. While (Jebara et al., 2004) introduced the so called probability product kernels which generalize a variety of discrepancy measures, its properties remain further study.

Motivated by above observations, our work focuses on a specialized probability kernel function which is a direct generalization of sample-based RKHS methods such as MMD. We focus on using plugin-estimator as the default estimator of the kernel function we defined, and illustrate that with the help of Bernstein polynomial techniques, we could analyze the bias and convergence bounds of these plugin-estimators. Our work thus connects the fields of discrete distribution-property estimation and kernel-based hypothesis test, which brings interesting possibilities.

## 2 NOTATION

We use bold symbol $\boldsymbol{p}, \boldsymbol{q} \in \mathbb{R}^k$ to represent a probability function over a discrete support with support size $k$, and $p_i, q_i$ represents the $i$th entry of $\boldsymbol{p}$ and $\boldsymbol{q}$. We use $\{v_1, v_2, ..., v_k\}, v_i \in \mathbb{R}^d$ to represent the support of $\boldsymbol{p}, \boldsymbol{q}$. $[k] := \{1, 2, 3..., k\}$ represents the set of indices of elements in $\{v_1, v_2, ..., v_k\}$. We use $\phi \circ (\boldsymbol{p}, \boldsymbol{q})$ to denote an element-wise function from $\mathbb{R}^k \times \mathbb{R}^k$ to $\mathbb{R}^k$, where $(\phi \circ (\boldsymbol{p}, \boldsymbol{q}))_i = \phi \circ (p_i, q_i)$ and $\phi \circ \boldsymbol{p}$ to denote an element-wise function from $\mathbb{R}^k$ to $\mathbb{R}^k$, where $(\phi \circ \boldsymbol{p})_i = \phi \circ p_i$. With a slight abuse of notation, we denote $\boldsymbol{p}^\rho$, $\boldsymbol{p} - \boldsymbol{q}$ as element-wise function defined above. We use $\mathrm{kernel}(\boldsymbol{p}, \boldsymbol{q})$ to denote kernel function which maps from $\mathbb{R}^k \times \mathbb{R}^k$ to real value $\mathbb{R}$. And $\mathrm{kernel}(x, y), x, y \in \mathbb{R}^d$ represents a kernel function from $\mathbb{R}^d \times \mathbb{R}^d$ to real value $\mathbb{R}$. We use $K$ to denote the gram matrix generated from $\mathrm{kernel}(x, y)$ on finite support $\{v_1, v_2, ..., v_k\}$, where $K_{ij} = \mathrm{kernel}(v_i, v_j)$. We use $\{x_1, x_2, ..., x_n\} \sim \boldsymbol{p}$ and $\{y_1, y_2, ..., y_n\} \sim \boldsymbol{q}$ to denote the samples from distribution $\boldsymbol{p}$ and $\boldsymbol{q}$, where $n$ is the sample size.

## 3 GENERALIZED PROBABILITY KERNEL

Probability kernel function, defined as kernel function between distributions instead of samples, is a natural extension of the idea of kernel function in sample space.

**Definition 1.** *Given distribution $\boldsymbol{p}$ and $\boldsymbol{q}$ belongs to a family of discrete distribution with the same finite support $\{v_1, v_2, ..., v_k\}, v_i \in \mathbb{R}^d$, where $k$ is the support size, we define the probability kernel function as $\mathrm{PK}(\boldsymbol{p}, \boldsymbol{q})$, which is a kernel function maps from $\mathbb{R}^k \times \mathbb{R}^k$ to real value $\mathbb{R}$.*

Many discrepancy measures, such as MMD, can serve as probability kernel functions, but people usually don't use the term of probability kernel function when describing them. The reason is that for most of the time, we only consider a limited number of distributions, and do not need or have the resources to navigate through all the distributions within the family. For example, when looking into the two-sample problem, we usually assume two samples $\{x_1, x_2, ..., x_n\} \in \mathbb{R}^d$ and $\{y_1, y_2, ..., y_n\} \in \mathbb{R}^d$ are i.i.d drawn from two distributions $\boldsymbol{p}$ and $\boldsymbol{q}$, and use the discrepancy measure $\mathrm{MMD}[\mathcal{F}, \boldsymbol{p}, \boldsymbol{q}]$ to determine if $\boldsymbol{p}$ and $\boldsymbol{q}$ are indistinguishable in the RKHS $\mathcal{F}$. We do not consider all other distributions in $\mathcal{F}$ that is irrelevant to our samples! So far the idea of kernel function between distributions is in practice not so much useful, however, here in this paper, we propose, when considering the plugin-estimator of many of the existing discrepancy measures, it is beneficial to view them as probability kernel functions.

### 3.1 DEFINATION OF GENERALIZED PROBABILITY KERNEL

**Definition 2** (Generalized probability kernel)**.** *Given the family $\mathcal{S}$ of discrete distribution on support $\{v_1, v_2, .., v_k\}$ where $v_i \in \mathbb{R}^d$. Let $\mathcal{F}$ be a unit ball in a universal-RKHS $\mathcal{H}$ with associated continuous kernel $\mathrm{RK}(x, y)$, where for any $x \in \mathbb{R}^d$ and $y \in \mathbb{R}^d$, $\mathrm{RK}(x, y)$ maps from $\mathbb{R}^d \times \mathbb{R}^d$ to $\mathbb{R}$. We denote gram matrix $K_{ij} = \mathrm{RK}(v_i, v_j)$.*

*The generalized probability kernel function on distribution $\boldsymbol{p}, \boldsymbol{q} \in \mathcal{S}$ is $\mathrm{GPK}_{\mathcal{F}, \phi}(\boldsymbol{p}, \boldsymbol{q}) = \phi \circ (\boldsymbol{p}, \boldsymbol{q}) \, K \phi \circ (\boldsymbol{q}, \boldsymbol{p})^T = \sum_{i \in [k]} \sum_{j \in [k]} \phi \circ (p_i, q_i) K_{ij} \phi \circ (q_j, p_j)$*

*where $\phi \circ (\boldsymbol{p}, \boldsymbol{q})$ is an element-wise mapping function on discrete distribution $\boldsymbol{p}, \boldsymbol{q} \in \mathcal{S}$, which maps from $\mathbb{R}^k \times \mathbb{R}^k$ to $\mathbb{R}^k$,*

Obviously, under this definition, the GPK is a symmetric probability kernel function where $\mathrm{GPK}_{\mathcal{F}, \phi}(\boldsymbol{p}, \boldsymbol{q}) = \mathrm{GPK}_{\mathcal{F}, \phi}(\boldsymbol{q}, \boldsymbol{p})$

Mapping function $\phi$ represent a great amount of possibilities. For most cases, we need to narrow down the region and equipped it with some convenient properties so that the GPK measure could be useful.

One example is for the measurement of discrepancy, where we want $\mathrm{GPK}_{\mathcal{F}, \phi}(\boldsymbol{p}, \boldsymbol{q}) = 0$ if and only if $\boldsymbol{p} = \boldsymbol{q}$.

**Definition 3** (discrepancy probability kernel)**.** *Let $\mathcal{S}$ be a family of discrete distribution $\boldsymbol{p} \in \mathcal{S}$ on support $\{v_1, v_2, ..., v_k\}$. A discrepancy probability kernel is a kernel function $\mathrm{PK}(\boldsymbol{p}, \boldsymbol{q})$ that $\mathrm{PK}(\boldsymbol{p}, \boldsymbol{q}) = 0$ if and only if $\boldsymbol{p} = \boldsymbol{q}$*

**Theorem 1.** $\mathrm{GPK}_{\mathcal{F},\phi}(\boldsymbol{p},\boldsymbol{q})$ *with the mapping function $\phi$ that satisfies:*

*1. symmetric or antisymmetric with respect to $\boldsymbol{p}$ and $\boldsymbol{q}$: $\phi \circ (\boldsymbol{p},\boldsymbol{q}) = \phi \circ (\boldsymbol{q},\boldsymbol{p})$ or $\phi \circ (\boldsymbol{p},\boldsymbol{q}) = -\phi \circ (\boldsymbol{q},\boldsymbol{p})$*

*2. $\|\phi \circ (\boldsymbol{p},\boldsymbol{q})\|_2 = \|\phi \circ (\boldsymbol{q},\boldsymbol{p})\|_2 = 0$ if and only if $\boldsymbol{p} = \boldsymbol{q}$, where $\|\cdot\|_2$ represents L2 norm.*

*is a discrepancy probability kernel.*

*Proof.*

$$\mathrm{GPK}_{\mathcal{F},\phi}(\boldsymbol{p},\boldsymbol{q}) = \sum_{i \in [k]} \sum_{j \in [k]} \phi \circ (p_i, q_i) K_{ij} \phi \circ (q_j, p_j)$$

$$= \phi \circ (\boldsymbol{p},\boldsymbol{q})\, K \phi \circ (\boldsymbol{q},\boldsymbol{p})^T = \pm \phi \circ (\boldsymbol{p},\boldsymbol{q})\, K \phi \circ (\boldsymbol{p},\boldsymbol{q})^T$$

$$= \pm \boldsymbol{v} K \boldsymbol{v}^T$$

$K$ is a semipositive definite matrix, thus by definition of positive definite matrix, $\boldsymbol{v} K \boldsymbol{v}^T \geq 0$, where equality holds if and only if $\boldsymbol{v} = \boldsymbol{0}$, and since $\boldsymbol{v} = \phi \circ (\boldsymbol{p},\boldsymbol{q})$, this condition further means $\phi \circ (\boldsymbol{p},\boldsymbol{q}) = 0$, which holds if and only if $\boldsymbol{p} = \boldsymbol{q}$.

$\square$

Another example is the polynomial GPK, which is our main focus of this paper. Such a subclass of GPK is interesting since we can build unbiased estimators of it using techniques of Bernstein polynomial in (Qian et al., 2011). As we will show in section 5., we also have analyzable convergence bounds for the resulting unbiased estimators, illustrating its potential usage for applications such as two-sample test.

**Definition 4** (polynomial GPK). *The polynomial GPK is the subset of GPK that equipped with the mapping function $\phi$ that is polynomial in $\boldsymbol{p}$ and $\boldsymbol{q}$: $\phi \circ (\boldsymbol{p},\boldsymbol{q}) = \sum_{l=0}^{o} \sum_{s=0}^{o} \alpha_{l,s} \boldsymbol{p}^l \boldsymbol{q}^s$ where $o \in \mathbb{Z}$ is the degree of the polynomial, and $a_{l,s} \in \mathbb{R}$ is the coefficient*

Below we give some examples of polynomial GPK, which include MMD proposed in (Gretton et al., 2012), and the newly proposed power-MMD in this paper, which is a natural extension of MMD, from the view point of probability kernels.

### 3.1.1 EXAMPLE 1: MMD AS MEMBER OF POLYNOMIAL GPK

Given discrete distribution $\boldsymbol{p}, \boldsymbol{q}$ with support $\{v_1, v_2, ..., v_k\}$, we can rewrite MMD with distribution probability function $p_i, q_i$:

$$\mathrm{MMD}^2_{\mathcal{F}}(\boldsymbol{p},\boldsymbol{q}) = \|\mathbf{E}_{x \sim \boldsymbol{p}} f(x) - \mathbf{E}_{x' \sim \boldsymbol{q}} f(x')\|^2_{\mathcal{H}}$$

$$= \left\| \sum_{i \in [k]} f(v_i) p_i - \sum_{i \in [k]} f(v_i) q_i \right\|^2_{\mathcal{H}} = \left\| \sum_{i \in [k]} f(v_i) p_i - f(v_i) q_i \right\|^2_{\mathcal{H}}$$

$$= \sum_{i \in [k]} \sum_{j \in [k]} (p_i - q_i) f(v_i) f(v_j)(p_j - q_j) = \sum_{i \in [k]} \sum_{j \in [k]} (p_i - q_i) K_{ij} (p_j - q_j)$$

$$= -\mathrm{GPK}_{\mathcal{F},\phi_l}(\boldsymbol{p},\boldsymbol{q})$$

Where $\phi_l \circ (\boldsymbol{p},\boldsymbol{q}) = \boldsymbol{p} - \boldsymbol{q}$, $\mathcal{H}$ is the RKHS defined in MMD literature, and $f$ is the function that maps $v_i$ to $\mathcal{H}$.

$\mathrm{GPK}_{\mathcal{F},\phi_l}(\boldsymbol{p},\boldsymbol{q})$ is a special case of polynomial GPK where $\alpha_{1,0} = 1$, $\alpha_{0,1} = -1$, and all other coefficients are 0.

### 3.1.2 EXAMPLE 2: PRODUCT GPK AS MEMBERS OF POLYNOMIAL GPK

**Definition 5** (product GPK). *Let $\boldsymbol{p}$ and $\boldsymbol{q}$ be probability distributions on support $\{v_1, v_2, ..., v_k\}$, and $l \in \mathbb{Z}$ be nonnegative integer. The product GPK is a subset of polynomial GPK where $\alpha_{l,0} = 1$, and all other coefficients are 0. the corresponding mapping function is: $\phi(\boldsymbol{p},\boldsymbol{q}) = \boldsymbol{p}^l$*

The probability product kernel as in (Jebara et al., 2004) is a special case of product GPK where $K$ is a identity matrix.

### 3.1.3 EXAMPLE 3: POWER-MMD AS MEMBERS OF POLYNOMIAL GPK

Another interesting subset of polynomial GPK is the one extends MMD case into a power form and we denote it as power-MMD:

**Definition 6** (power-MMD). *Let $p$ and $q$ be probability distributions on support $\{v_1, v_2, ..., v_k\}$ and $\rho \in \mathbb{Z}$ be a positive integer. then the power-MMD is a subset of polynomial GPK where $\alpha_{\rho,0} = 1$, $\alpha_{0,\rho} = -1$, and all other coefficients are 0. the corresponding mapping function is: $\phi(\boldsymbol{p}, \boldsymbol{q}) = \boldsymbol{p}^\rho - \boldsymbol{q}^\rho$*

Apparently, MMD is a special case of power-MMD where $\rho = 1$, and power-MMD satisfies the requirement in Theorem 1, thus has the potential usage of discrepancy measure. In section 5., we will show that power-MMD has unbiased estimator with analyzable convergence bounds thus could be used for two-sample test.

### 3.2 DISCUSSION OF GPK IN DISCRETE SETTING

As one may easily notice, the definition of GPK includes a gram matrix generated by the kernel function $\mathsf{RK}(v_i, v_j)$ which measures the discrepancy between $v_i, v_j \in \{v_1, v_2, .., v_k\}$. While considering the cases of categorical distribution, values of discrete variables does not relate to any notion of distance, this raises the question: how the introduced gram matrix will be beneficial in any cases?

The answer is twofold: 1. Many natural processes produce discrete distributions where there possibly exists a similarity measure in values which imply the similarity in frequencies of occurrence(probability values). For example, in the field of natural language process(NLP), one may treat words as atomic units with no notion of similarity between words, as these are represented as indices in a vocabulary. However, given large number of training samples, similarity measure between words could be made possible using techniques such as words2vec(Mikolov et al., 2013). Such techniques generally result in better performance and have become the important preprocessing techniques for NLP tasks(Goodfellow et al., 2016). 2. As there are cases where the values of discrete variables are totally irrelevant, or people may use kernel function $\mathsf{RK}(v_i, v_j)$ which doesn't correctly imply the similarity in probability values, the GPK framework may still capture the similarity between distributions. One example is the case of MMD, which is, as we discussed above, an element of GPK family. As proved in (Gretton et al., 2012), MMD is a distribution free measurement between two samples, which means no matter what kind of $\boldsymbol{p}, \boldsymbol{q}$ and kernel$(x, y)$, we have, the $\mathrm{MMD}_{\mathcal{F}}^2(\boldsymbol{p}, \boldsymbol{q})$ measure will be 0 if and only if $\boldsymbol{p} = \boldsymbol{q}$. However, the bad choice of kernel function does have a negative effect on convergence bounds of the empirical estimator proposed in (Gretton et al., 2012), and will influence the results of two-sample test. For this reason, we mainly focus on dataset with known relativity measures in our experiment section.

## 4 PLUGIN-ESTIMATOR FOR GPK

So far we have defined the GPK and discussed some subsets of GPK with potential usage of two-sample test. Next we discuss how to build an estimator, given a member of GPK. In this section, we propose the plugin-estimator, which based on the count of occurrence of each value $v_i \in \{v_1, v_2, ..., v_k\}$ in samples $\{x_1, x_2, ..., x_n\} \in \boldsymbol{p}$ or $\boldsymbol{q}$. We illustrate that by doing so, the techniques of Bernstein polynomial in (Qian et al., 2011) could be used to help building unbiased estimators for any members of polynomial GPK. Furthermore, we provide analyzable convergence bounds of these estimators.

We begin with the definition of plugin-estimators:

**Definition 7.** *Suppose we have i.i.d samples of distribution $\boldsymbol{p}$ as $X_{n_1} := \{x_1, x_2, ..., x_{n_1}\} \sim \boldsymbol{p}$ and $X_{n_2} := \{x_{n_1+1}, x_{n_1+2}, ..., x_{n_1+n_2}\} \sim \boldsymbol{p}$. And also the i.i.d samples of distribution $\boldsymbol{q}$ as $Y_{m_1} := \{y_1, y_2, ..., y_{m_1}\} \sim \boldsymbol{q}$ and $Y_{m_2} := \{y_{m_1+1}, y_{m_1+2}, ..., y_{m_1+m_2}\} \sim \boldsymbol{q}$.*

Let $N_i^{(n_1)}$ denotes the number of occurrence of value $v_i \in \{v_1, v_2, ..., v_k\}$ in sample $X_{n_1}$, and $S_{i,n_1} := (N_i^{(n_1)}, n_1)$ denotes the collection of $N_i^{(n_1)}$ and $n_1$. The same follows for $X_{n_2}$, $Y_{m_1}$ and $Y_{m_2}$

We define the plugin-estimator of $\text{GPK}_{\mathcal{F}, \phi}(\boldsymbol{p}, \boldsymbol{q})$ as

$$\text{GPK}_E[\mathcal{F}, \phi, X, Y] = \sum_{i \in [k]} \sum_{j \in [k]} f_\phi\left(S_{i,n_1}, S_{i,m_1}\right) K_{ij} f_\phi\left(S_{j,m_2}, S_{j,n_2}\right)$$

where $f_\phi$ is a function related to function $\phi$, and $K$ is the gram matrix brought by $\mathcal{F}$.

Here our setting is different from the unbiased estimator $\text{MMD}_u^2$ of (Gretton et al., 2012), where in their setting $X_{n_1}$, $X_{n_2}$ represent the same sample from $\boldsymbol{p}$ and so do for $Y_{m_1}$, $Y_{m_2}$ from $\boldsymbol{q}$. Instead, we are using the same setting as the linear time statistic $\text{MMD}_l^2$ proposed in (Gretton et al., 2012). Another way of viewing this is that for our setting, given two samples $\{x_1, x_2, ..., x_n\}$, $\{y_1, y_2, ..., y_n\}$ from $\boldsymbol{p}$ and $\boldsymbol{q}$, we depart each sample of $x$ and $y$ into two parts, yielding 4 different samples with size $n_1, n_2, m_1, m_2$, and then calculate the empirical frequencies for plugin-estimator defined above.

## 4.1 POLYNOMIAL GPK WITH UNBIASED PLUGIN-ESTIMATORS

One of our main contributions of this paper is the proposal that we can always find an unbiased plugin-estimator for any members in polynomial GPK family. The basic idea is that we can analyze the expectation of plugin-estimators through Bernstein polynomial, and use the existing results of (Qian et al., 2011) to build the unbiased plugin estimators.

**Theorem 2.** *Denote*

$$g_j(k, n) := \begin{cases} g_j(k, n) = \left( \begin{array}{c} k \\ j \end{array} \right) \left( \begin{array}{c} n \\ j \end{array} \right)^{-1}, & \text{for } j \leq k \\ 0, & \text{for } j > k \end{cases}$$

*Then any member of polynomial* $\text{GPK}[\mathcal{F}, \phi, \boldsymbol{p}, \boldsymbol{q}]$ *equipped with polynomial mapping function* $\phi(\boldsymbol{p}, \boldsymbol{q}) = \sum_{l=0}^{o} \sum_{s=0}^{o} \alpha_{l,s} \boldsymbol{p}^l \boldsymbol{q}^s$ *of degree* $o \in \mathbb{Z}$, *has an unbiased plugin-estimator with mapping function* $f_\phi$ *to be:*

$$f_\phi(S_{i,n_1}, S_{i,m_1}) = \sum_{l=0}^{o} \sum_{s=0}^{o} \alpha_{l,s} g_l(N_i^{(n_1)}, n_1) g_s(N_i^{(m_1)}, m_1)$$

*Proof.* The basic idea is directly using the result of Bernstein polynomial in (Qian et al., 2011) to build unbiased estimators. We put our formal proof in appendix $\qquad \square$

For notation simplicity, we define the plugin-estimator discussed above to be the default-plugin-estimator for polynomial GPK:

**Definition 8** (default-plugin-estimator for polynomial GPK). *The plugin-estimator defined in Theorem 2 is the default-plugin-estimator for polynomial GPK.*

*This plugin-estimator, according to Theorem 2, is an unbiased estimator*

## 4.2 DEVIATION BOUND OF PLUGIN-ESTIMATORS

Another topic about plugin-estimator is its deviation bound. We directly use the McDiamid's inequality to derive the default-plugin-estimator for polynomial GPK:

**Theorem 3.** *The default-plugin-estimator of* $\text{GPK}[\mathcal{F}, \phi, \boldsymbol{p}, \boldsymbol{q}]$ *equipped with polynomial mapping function* $\phi(\boldsymbol{p}, \boldsymbol{q}) = \sum_{l=0}^{o} \sum_{s=0}^{o} \alpha_{l,s} \boldsymbol{p}^l \boldsymbol{q}^s$ *of degree* $o \in \mathbb{Z}$

*has the convergence bound:*

$$\forall a > 0, \Pr(|\text{GPK}_E[\mathcal{F}, \phi, X, Y] - \mathbb{E}[\text{GPK}_E[\mathcal{F}, \phi, X, Y]]| \geq a) \leq 2e^{-\frac{2a^2}{Z}}$$

*where*

$$Z = \left( \left( n_1 \left( \tau_{n_1,m_1}^{(1)} \right)^2 + m_1 \left( \tau_{n_1,m_1}^{(2)} \right)^2 \right) \Phi_{m_2,n_2}^2 + \left( m_2 \left( \tau_{m_2,n_2}^{(1)} \right)^2 + n_2 \left( \tau_{m_2,n_2}^{(2)} \right)^2 \right) \Phi_{n_1,m_1}^2 \right) K_{max}^2$$

$$\Phi_{n,m} = \sum_{i\in[k]} \left| \sum_{l=0}^{o} \sum_{s=0}^{o} \alpha_{l,s} g_l(N_i^{(n)}, n) g_s(N_i^{(m)}, m) \right|$$

$$\tau_{n,m}^{(1)} = \sup_{i\in[k]} \left( \sum_{l=0}^{o} \sum_{s=0}^{o} \frac{l}{N_i^{(n)}} \cdot |\alpha_{l,s}| \cdot g_l(N_i^{(n)}, n) g_s(N_i^{(m)}, m) \right)$$

$$\tau_{n,m}^{(2)} = \sup_{i\in[k]} \left( \sum_{l=0}^{o} \sum_{s=0}^{o} \frac{s}{N_i^{(m)}} \cdot |\alpha_{l,s}| \cdot g_l(N_i^{(n)}, n) g_s(N_i^{(m)}, m) \right)$$

$K_{max}$ *is the largest value of entries in* $K$

*Proof.* The basic idea is to use the McDiamid's inequality, and we put our formal proof into the appendix. $\qquad\square$

## 5 EXAMPLE: POWER-MMD AS A NATURAL EXTENSION TO MMD FROM GPK VIEWPOINT

In this section, we mainly discuss power-MMD as defined in 3.1.3. We analyze the bias and convergence bound of its plugin-estimators using the techniques we introduced so far, illustrating that such a natural extension to MMD from GPK viewpoint could be beneficial for two-sample test.

### 5.1 PLUGIN-ESTIMATORS OF POWER-MMD

As we already discussed in section 4.1.3, power-MMD is a subset of polynomial GPK. According to Theorem 2, any member $\text{GPK}_{\mathcal{F},\phi_\rho}(\boldsymbol{p},\boldsymbol{q})$ in power-MMD has a default-plugin-estimator with the mapping function $f_\phi(S_{i,n_1}, S_{i,m_1}) = g_\rho(N_i, n_1) - g_\rho(M_i, m_1)$

**Remark 3.1.** *When* $\rho = 1$*, the power-MMD return to the original MMD case. Remarkably, the default-plugin-estimator of this case is equivalent to the linear time statistic* $\text{MMD}_l^2$ *proposed in (Gretton et al., 2012):* $\text{GPK}_E[\mathcal{F},\phi_l, X, Y] = \text{MMD}_l^2[\mathcal{F}, X, Y]$ *For details of the derivation, see appendix*

### 5.2 DEVIATION BOUND OF PLUGIN-ESTIMATORS OF POWER-MMD

**Corollary 3.1.** *Denote* $\tau_n = \sup_{i\in[k]} \left( \frac{\rho}{N_i^{(n)}} g_\rho(N_i^{(n)}, n) \right)$

*The default-plugin-estimator of power-MMD* $\text{GPK}_{\mathcal{F},\phi_\rho}(\boldsymbol{p},\boldsymbol{q})$ *has uniform convergence bound defined in Theorem 3. with* $\tau_{n,m}^{(1)} = \tau_n$ *and* $\tau_{n,m}^{(2)} = \tau_m$

**Corollary 3.2.** *Consider the case where* $n_1 = n_2 = m_1 = m_2 = n$ *The default-plugin-estimator of power-MMD* $\text{GPK}_{\mathcal{F},\phi_\rho}(\boldsymbol{p},\boldsymbol{q})$ *has uniform convergence bound:*

$$\Pr(|\text{GPK}_E[\mathcal{F},\phi_\rho, X, Y] - \mathbb{E}[\text{GPK}_E[\mathcal{F},\phi_\rho, X, Y]]| \geq a) \leq 2e^{\frac{-na^2}{(\rho^2\Phi_{n_1,m_1}^2 + \rho^2\Phi_{m_2,n_2}^2)K_{max}^2}} \leq 2e^{\frac{-na^2}{8\rho^2 K_{max}^2}}$$

*Proof.* The first inequality above comes from:

$$\frac{\rho}{N_i} g_\rho(N_i, n) = \frac{\rho(N_i-1)(N_i-2)...(N_i-\rho+1)}{n(n-1)...(n-\rho+1)} \leq \frac{\rho}{n} \frac{N_i^{\rho-1}}{n^{\rho-1}} \leq \frac{\rho}{n}$$

where $sup_{i\in[k]}\left(\frac{\rho}{N_i}g_\rho(N_i,n)\right)=\frac{\rho}{n}$ only stands for extreme case such that there exist $N_i=n$, i.e. all the samples belongs to the same value $v_i\in\{v_1,v_2,...,v_k\}$. And the second inequality above comes from:

$$\Phi_{n,m}=\sum_{i\in[k]}\left|g_\rho(N_i^{(n)},n)-g_\rho(N_i^{(m)},m)\right|\leq\sum_{i\in[k]}\left|g_\rho(N_i^{(n)},n)\right|+\sum_{i\in[k]}\left|g_\rho(N_i^{(m)},m)\right|$$

$$\leq\sum_{i\in[k]}\left(\frac{N_i^{(n)}}{n}\right)^\rho+\sum_{i\in[k]}\left(\frac{N_i^{(m)}}{m}\right)^\rho\leq\left(\sum_{i\in[k]}\frac{N_i^{(n)}}{n}\right)^\rho+\left(\sum_{i\in[k]}\frac{N_i^{(m)}}{m}\right)^\rho=2$$

$\square$

**Remark 3.2.** *recall in (Gretton et al., 2012), the deviation bound for linear time estimator* $\mathrm{MMD}_l^2[\mathcal{F},X,Y]$ *is*

$$\forall a>0,\Pr(\left|\mathrm{MMD}_l^2[\mathcal{F},X,Y]-\mathbb{E}[\mathrm{MMD}_l^2[\mathcal{F},X,Y]]\right|\geq a)\leq 2e^{\frac{-na^2}{8K_{max}^2}}$$

*Interestingly, this bound is the same as the case $\rho=1$ in Corollary 3.2. Note that according to section 6.1, the default-plugin-estimator of power-MMD with $\rho=1$ is actually in equivalent to $\mathrm{MMD}_l^2$ case in (Gretton et al., 2012). Our bound generalize the bound in (Gretton et al., 2012) and provide a tighter version. Note that the bounds for special case of $\rho=1$ has simpler derivation, and the reader may refer to appendix for more details.*

### 5.3 TWO-SAMPLE TEST USING POWER-MMD

**Corollary 3.3.** *A hypothesis test of level $\alpha$ for the null hypothesis $\boldsymbol{p}=\boldsymbol{q}$ has the acceptance region* $\left|\frac{\mathrm{GPK}_E[\mathcal{F},\phi_\rho,X,Y]}{\sqrt{Z}}\right|<\sqrt{\frac{1}{2}log((\frac{\alpha}{2})^{-1})}$ *Where $Z$ is defined in Corollary 3.1*

The two-sample test for power-MMD then follows this procedure: 1. calculate $v=\left|\frac{\mathrm{GPK}_E[\mathcal{F},\phi_\rho,X,Y]}{\sqrt{Z}}\right|$. 2. check if $v<\sqrt{\frac{1}{2}log((\frac{\alpha}{2})^{-1})}$, if so, accept the null hypothesis, otherwise reject the null hypothesis.

Next we analyze the performance of our proposed two-sample test under two cases: $\rho=1$ and $\rho>1$

#### 5.3.1 $\rho=1$ CASE

For $\rho=1$ case, since $\mathrm{GPK}_E[\mathcal{F},\phi_1,X,Y]$ is equivalent to $\mathrm{MMD}_l^2[\mathcal{F},\phi_1,X,Y]$, the only difference between our proposal and that of Gretton et al. (2012) is the convergence bound. According to Remark 3.2, we provide a tighter bound for the test statistic, thus we will certainly have a better performance using power-MMD.

#### 5.3.2 $\rho>1$ CASE

We need to answer two questions for the $\rho>1$ case: 1. when applying power-MMD in practice, is the proposed statistics numerical stable? 2. will the performance of two-sample test gets better when $\rho$ gets larger?

For the question of numerical stability, since $g_\rho(N_i,n)\leq\left(\frac{N_i}{n}\right)^\rho$, the term will exponentially decrease with the increase of $\rho$. This effect will cause numerical problem when $N_i\ll n$ and $\rho$ is large. One solution is to find an upper-bound of $\left|\frac{\mathrm{GPK}_E[\mathcal{F},\phi_\rho,X,Y]}{\sqrt{Z}}\right|$ which is numerical stable.

**Corollary 3.4.** *Consider the simplest case where $n_1=n_2=m_1=m_2=n$. Define*

$$C_N:=\{N_1^{(n_1)},N_2^{(n_1)},...,N_k^{(n_1)},N_1^{(n_2)},N_2^{(n_2)},...,N_k^{(n_2)},N_1^{(m_1)},N_2^{(m_1)},...,$$
$$N_k^{(m_1)},N_1^{(m_2)},N_2^{(m_2)},...,N_k^{(m_2)}\}$$

*to be the set of all counts of occurrence in the four samples $X_{n_1}, X_{n_2}, X_{m_1}, X_{m_2}$. Denote $S_N = \sup_{N_i \in C_N}(N_i)$ to be the maximium value in the set $C_N$ We have:*

$$\left| \frac{\mathrm{GPK}_E[\mathcal{F}, \phi_\rho, X, Y]}{\sqrt{Z}} \right| \leq \left| \frac{S_N \cdot \mathrm{GPK}'_E}{K_{max}\rho\sqrt{2n} \cdot \Phi'} \right|$$

*where*

$$\mathrm{GPK}'_E := \sum_{i,j \in [k]} \left( g_\rho(N_i^{(n_1)}, n) - g_\rho(N_i^{(m_1)}, S_N) \right) K_{ij} \left( g_\rho(N_j^{(n_2)}, S_N) - g_\rho(N_j^{(m_2)}, S_N) \right)$$

*and*

$$\Phi' = \sum_{i \in [k]} \left| g_\rho(N_i^{(n)}, S_N) - g_\rho(N_i^{(m)}, S_N) \right|$$

For cases when $N_i$ are not far less from $S_N$, $\mathrm{GPK}'_E$ will be much more numerical stable than $\mathrm{GPK}_E[\mathcal{F}, \phi_\rho, X, Y]$.

To answer the question related to the performance of two-sample test when $\rho$ get larger, we need to analyze the case when $\boldsymbol{p} \neq \boldsymbol{q}$, if $\left| \frac{\mathrm{GPK}_E[\mathcal{F}, \phi_\rho, X, Y]}{\sqrt{Z}} \right|$ increase with the increase of $\rho$. Unfortunately, there is no clear answer to this.

## 6  SUMMARY

To summarize, we introduce the framework of generalized probability kernel(GPK). While GPK represents a large family of probability kernels, we focus on polynomial GPK since all members of such subset of GPK have unbiased plugin-estimators. Remarkably, a natural extension of MMD from the viewpoint of polynomial GPK, which we call power-MMD, could be used for two-sample test. Theoretical study shows that for $\rho = 1$ case, power-MMD outperforms linear time MMD proposed in Gretton et al. (2012), and the performance of $\rho > 1$ case is left for future work. For members of GPK which do not belong to polynomial GPK, it is not easy to design an unbiased estimators. However, bais reduction techniqes proposed in (Yi et al., 2018) and (Yi & Alon, 2020) could be used, and we still have the chance to apply two-sample test with the resulting estimators. Such a possibility is also left for future work.

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

# A APPENDIX

## A.1 BERNSTEIN POLYNOMIAL

Drawing i.i.d. samples $Y^m$ from any distribution $\boldsymbol{p}$ the expected value of the empirical estimator for a distribution property is

$$\mathbb{E}\left[\hat{H}_{\mathrm{E}}\left(Y^m\right)\right] = \sum_{i\in[k]} \underset{M_i\sim\mathrm{bin}(m,p_i)}{\mathbb{E}} \left[h\left(\frac{M_i}{m}\right)\right]$$

Note that for any function $f, m \in \mathbb{N}$, and $x \in [0,1]$, the degree-m Bernstein polynomial of $f$ is

$$B_m(f,x) := \sum_{j=0}^{m} f\left(\frac{j}{m}\right) \binom{m}{j} x^j (1-x)^{m-j}$$

Therefore, we can express the expectation of the empirical property estimator as

$$\underset{Y^m\sim p}{\mathbb{E}} \left[\hat{H}_{\mathrm{E}}\left(Y^m\right)\right] = \sum_{i\in[k]} B_m\left(h, p_i\right)$$

## A.2 PROOF OF THEOREM 2

*Proof.* Recall the definition of polynomial GPK:

$$\mathrm{GPK}(\mathcal{F},\phi,\boldsymbol{p},\boldsymbol{q}) = \sum_{i\in[k]}\sum_{j\in[k]} \phi(p_i,q_i)K_{i,j}\phi(q_j,p_j) = \sum_{i\in[k]}\sum_{j\in[k]} K_{i,j} \sum_{l,s,r,t=0}^{o} \alpha_{l,s}\alpha_{r,t}p_i^l q_i^s p_j^t q_j^r$$

recall in (Qian et al., 2011)

$$x^j = \sum_{k=j}^{n} g_j(k,n)b_{k,n}(p_i) = \mathbb{E}_{k\sim\mathrm{bin}(p_i,n)}g_j(k,n)$$

where $g_j(k,n)$ is defined in the beginning of the theorem

$$\mathrm{GPK}(\mathcal{F},\phi,\boldsymbol{p},\boldsymbol{q}) = \sum_{i\in[k]}\sum_{j\in[k]} K_{i,j} \sum_{l,s,r,t=0}^{o} \alpha_{l,s}\alpha_{r,t}p_i^l q_i^s p_j^t q_j^r$$

$$= \sum_{i\in[k]}\sum_{j\in[k]} K_{i,j} \sum_{l,s,r,t=0}^{o} \alpha_{l,s}\alpha_{r,t}\mathbb{E}_{N_i^{(n_1)}\sim\mathrm{bin}(p_i,n_1)}g_l(N_i,n_1)$$

$$\mathbb{E}_{N_i^{(m_1)}\sim\mathrm{bin}(q_i,m_1)}g_s(N_i^{(m_1)},n_2)\mathbb{E}_{N_j^{(n_2)}\sim\mathrm{bin}(p_j,n_2)}g_t(N_j^{(n_2)},n_2)\mathbb{E}_{N_j^{(m_2)}\sim\mathrm{bin}(q_j,m_2)}g_r(N_j^{(m_2)},m_2)$$

$$= \mathbb{E}\left[\sum_{i\in[k]}\sum_{j\in[k]} K_{i,j} \sum_{l,s,r,t=0}^{o} \alpha_{l,s}\alpha_{r,t}g_l(N_i^{(n_1)},n_1)g_s(N_i^{(m_1)},m_1)g_t(N_j^{(n_2)},n_2)g_r(N_j^{(m_2)},m_2)\right]$$

$$= \mathbb{E}\left[\mathrm{GPK}_E(\mathcal{F},\phi,X,Y)\right]$$

$\square$

## A.3 PROOF OF THEOREM 3.

**Lemma 4.** *Let $S_{N_i,n_1} := (N_i^{(n_1)},n_1)$ denotes the collection of $N_i^{(n_1)}$ and $n_1$. The same follows for $X_{n_2}, Y_{m_1}$ and $Y_{m_2}$. Also for notation simplicity, let $S_{i,n_1} := S_{N_i,n_1}$*

*For the plugin-estimator*

$$\mathrm{GPK}_E[\mathcal{F},\phi,X,Y] = \sum_{i\in[k]}\sum_{j\in[k]} f_\phi\left(S_{N_i,n_1}, S_{N_i,m_1}\right) K_{ij} f_\phi\left(S_{N_j,m_2}, S_{N_j,n_2}\right)$$

*with mapping function $f_\phi(S_{N_i,n_1}, S_{N_i,m_1})$ having the following properties:*

- $f_\phi(S_{N_i,n_1}, S_{N_i,m_1})$ *is a monotonic function related to* $N_i^{(n_1)}$ *and* $N_i^{(m_1)}$:

$$\forall N_i > N_i' \text{ or } \forall N_i < N_i' : f_\phi(S_{N_i,n_1}, S_{N_i,m_1}) > f_\phi(S_{N_i',n_1}, S_{N_i,m_1})$$

*The same follows for* $N_i^{(m_1)}$

- 

$$|f_\phi(S_{N_i\pm 1,n_1}, S_{N_i,m_1}) - f_\phi(S_{N_i,n_1}, S_{N_i,m_1})| \leq \tau_{n_1}$$

*and*

$$|f_\phi(S_{N_i,n_1}, S_{N_i\pm 1,m_1}) - f_\phi(S_{N_i,n_1}, S_{N_i,m_1})| \leq \tau_{m_1}$$

*where* $\tau_{n_1}$ *is a constant related to sample size* $n_1$, $\tau_{m_1}$ *is a constant related to sample size* $m_1$, *the same follows for* $\tau_{n_2}$ *and* $\tau_{m_2}$

*We have:*

$$\forall a > 0, \Pr(|\mathrm{GPK}_E[\mathcal{F}, \phi, X, Y] - \mathbb{E}[\mathrm{GPK}_E[\mathcal{F}, \phi, X, Y]]| \geq a) \leq 2e^{\frac{-2a^2}{Z}}$$

*where*

$$Z = \left(\left(n_1\tau_{n_1}^2 + n_2\tau_{n_2}^2\right)\Phi_2^2 + \left(m_1\tau_{m_1}^2 + m_2\tau_{m_2}^2\right)\Phi_1^2\right)K_{max}^2$$

$$\Phi_1 = \sum_{j\in[k]}\left|f_\phi(S_{N_j,n_1}, S_{N_j,m_1})\right|, \Phi_2 = \sum_{j\in[k]}\left|f_\phi(S_{N_j,n_2}, S_{N_j,m_2})\right|$$

$K_{max}$ *is the largest value of entries in* $K$

*Proof.* recall McDiamid's inequality

**Theorem 5.** *Let* $Y_1, \ldots, Y_m$ *be independent random variables taking values in ranges* $R_1, \ldots, R_m$ *and let* $F : R_1 \times \ldots \times R_m \to C$ *with the property that if one freezes all but the* $w^{th}$ *co-ordinate of* $F(y_1, \ldots, y_m)$ *for some* $1 \leq w \leq m$, *then* $F$ *only fluctuates by most* $c_w > 0$, *thus* $\mid F(y_1, \ldots, y_{w-1}, y_w, y_{w+1}, \ldots, y_m) - F(y_1, \ldots, y_{w-1}, y_w', y_{w+1}, \ldots, y_m) \mid \leq c_w$ *for all* $y_j \in R_j$ *and* $y_w' \in R_w$ *for* $1 \leq j \leq m$

*Then for any* $a > 0$, *one has* $\Pr(|F(Y) - \mathbb{E}[F(Y)]| \geq a) \leq 2e^{-\frac{2a^2}{\Sigma_{i=1}^n c_i^2}}$

considering the plugin-estimator of GPK family:

$$\mathrm{GPK}_E[\mathcal{F}, \phi, X, Y] = \sum_{i\in[k]}\sum_{j\in[k]} f_\phi(S_{i,n_1}, S_{i,m_1}) K_{ij} f_\phi(S_{j,m_2}, S_{j,n_2})$$

Without loss of generality, we rewrite the function $f_\phi$ as:

$$f_\phi(S_{N_i,n_1}, S_{N_i,m_1}) = F(x_1, x_2, ..., x_s, ..., x_{n_1}, N_i^{(m_1)}, m_1) = F_{N_i}$$

Assume we freeze all but one element in $X_{n_1} := \{x_1, x_2, ..., x_s, ..., x_{n_1}\}$, and only $x_s$ is allowed to change its value.

obviously, no matter how this element change, it always lies in the finite set of support $\{v_1, v_2, ..., v_k\}$, without loss of generality, we assume $x_s$ changes its value from $v_i$ to $v_{ii}$, thus the corresponding count of occurrence $N_i$ changes to $N_i - 1$ and $N_{ii}$ changes to $N_{ii} + 1$

we have for $x_s \in X_{n_1}$

$$c_s = sup_{\underline{x}_s} \left| \mathbb{GPK}_E(x_1, x_2, ..., x_n) - \mathbb{GPK}_E(x_1, x_2, ..., \underline{x}_s, ..., x_n) \right|$$

$$= sup_{\underline{x}_s} \left| \sum_{i,j \in [k]} \left( F(x_1, x_2, ..., x_s, ..., x_{n_1}, N_i^{(m_1)}, m_1) - F(x_1, x_2, ..., \underline{x}_s, ..., x_{n_1}, N_i^{(m_1)}, m_1) \right) K_{ij} f_\phi \left( S_{j,m_2}, S_{j,n_2} \right) \right|$$

$$= sup_{i,ii \in [k]} \left| \sum_{j \in [k]} \left( (F_{N_i - 1} - F_{N_i}) K_{i,j} + (F_{N_{ii} + 1} - F_{N_{ii},1}) K_{ii,j} \right) f_\phi (S_{N_j, n_2}, S_{N_j, m_2}) \right|$$

$$\leq \left| \sum_{j \in [k]} \tau_{n_1}(-K_{i,j} + K_{ii,j}) f_\phi(S_{N_j, n_2}, S_{N_j, m_2}) \right| \leq |\tau_{n_1} K_{max}| \sum_{j \in [k]} \left| f_\phi(S_{N_j, n_2}, S_{N_j, m_2}) \right|$$

$$= \tau_{n_1} K_{max} \Phi_2$$

where $\Phi_2 = \sum_{j \in [k]} \left| f_\phi(S_{N_j, n_2}, S_{N_j, m_2}) \right|$

Note that $K_{i,j} := K_{ij}$

Similarly, for $x_s \in X_{n_2}$

$$c_s \leq \tau_{n_2} K_{max} \Phi_1$$

where $\Phi_1 = \sum_{j \in [k]} |\phi(p_{1j}, q_{1j})|$

for $x_s \in Y_{m_1}$

$$c_s \leq \tau_{m_1} K_{max} \Phi_2$$

for $x_s \in Y_{m_2}$

$$c_s \leq \tau_{m_2} K_{max} \Phi_1$$

thus according to McDiamid's inequality, we have

$$\Pr(|\mathbb{GPK}_E[\mathcal{F}, \phi, X, Y] - \mathbb{E}\left[ \mathbb{GPK}_E[\mathcal{F}, \phi, X, Y] \right]| \geq a) \leq 2e^{\frac{-2a^2}{\sum_{i=1}^{n_1 + n_2 + m_1 + m_2} c_i^2}}$$

We set

$$c_i = \tau_{n_1} K_{max} \Phi_2$$

for $x_i \in \{X_{n_1}\}$,

$$c_i = \tau_{m_1} K_{max} \Phi_2$$

for $x_i \in \{Y_{m_1}\}$,

$$c_i = \tau_{n_2} K_{max} \Phi_1$$

for $x_i \in \{X_{n_2}\}$,

$$c_i = \tau_{m_2} K_{max} \Phi_1$$

for $x_i \in \{Y_{m_2}\}$

and get

$$\sum_{i=1}^{n_1 + n_2 + m_1 + m_2} c_i^2 = n_1 \cdot (\tau_{n_1} K_{max} \Phi_2)^2 + n_2 \cdot (\tau_{n_2} K_{max} \Phi_1)^2$$

$$+ m_1 \cdot (\tau_{m_1} K_{max} \Phi_2)^2 + m_2 \cdot (\tau_{m_2} (K_{max} \Phi_1)^2$$

$$= \left( (n_1 \tau_{n_1}^2 + m_1 \tau_{m_1}^2) \Phi_2^2 + (n_2 \tau_{n_2}^2 + m_2 \tau_{m_2}^2) \Phi_1^2 \right) K_{max}^2$$

$\square$

We are ready to proof theorem 3:

*Proof.* Define

$$
\begin{aligned}
Div_{n,l,s}^{(1)} &= |g_l(N_i^{(n)} + 1, n)g_s(N_i^{(m)}, m) - g_l(N_i^{(n)}, n)g_s(N_i^{(m)}, m)| \\
&= \left| \frac{(N_i + 1)N_i(N_i - 1)...(N_i - \rho + 2)}{n_1(n_1 - 1)...(n_1 - \rho + 1)} - \frac{N_i(N_i - 1)...(N_i - \rho + 1)}{n_1(n_1 - 1)...(n_1 - \rho + 1)} \right| \cdot \left| g_s(N_i^{(m)}, m) \right| \\
&= \left| \frac{l}{N_i^{(n)} - l + 1} g_l(N_i^{(n)}, n) \right| \cdot \left| g_s(N_i^{(m)}, m) \right| \\
&= \frac{l}{N_i^{(n)} - l + 1} g_l(N_i^{(n)}, n)g_s(N_i^{(m)}, m)
\end{aligned}
$$

$$
\begin{aligned}
Div_{n,l,s}^{(2)} &= |g_l(N_i^{(n)} - 1, n)g_s(N_i^{(m)}, m) - g_l(N_i^{(n)}, n)g_s(N_i^{(m)}, m)| \\
&= \left| \frac{(N_i - 1)(N_i - 2)...(N_i - \rho)}{n_1(n_1 - 1)...(n_1 - \rho + 1)} - \frac{N_i(N_i - 1)...(N_i - \rho + 1)}{n_1(n_1 - 1)...(n_1 - \rho + 1)} \right| \cdot \left| g_s(N_i^{(m)}, m) \right| \\
&= \left| \frac{l}{N_i^{(n)}} g_l(N_i^{(n)}, n) \right| \cdot \left| g_s(N_i^{(m)}, m) \right| \\
&= \frac{l}{N_i^{(n)}} g_l(N_i^{(n)}, n)g_s(N_i^{(m)}, m)
\end{aligned}
$$

$$
Div_{n,l,s} = \max(Div_{n,l,s}^{(1)}, Div_{n,l,s}^{(2)}) = \frac{l}{N_i^{(n)}} g_l(N_i^{(n)}, n)g_s(N_i^{(m)}, m)
$$

Recall the mapping function of default-plugin-estimator of polynomial GPK $f_\phi(S_{i,n}, S_{i,m}) := \sum_{l=0}^{o} \sum_{s=0}^{o} \alpha_{l,s} g_l(N_i^{(n)}, n)g_s(N_i^{(m)}, m)$

Apparently $f_\phi$ is a monotonic function with respect to $N_i$ and $M_i$, thus condition 1. for Theorem 3. is satisfied

Since we also have:

$$
\begin{aligned}
|f_\phi(S_{N_i^{(n)} \pm 1, n}, S_{N_i^{(m)}, m}) - f_\phi(S_{N_i, n_1}, S_{M_i, m_1})| &\leq \sum_{l=0}^{o} \sum_{s=0}^{o} |\alpha_{l,s} Div_{n,l,s}| \\
&= \sum_{l=0}^{o} \sum_{s=0}^{o} |\alpha_{l,s}| \frac{l}{N_i^{(n)}} g_l(N_i^{(n)}, n)g_s(N_i^{(m)}, m)
\end{aligned}
$$

thus condition 2. for Theorem 3. is satisfied

$\square$

## A.4 PROOF OF COROLLARY 3.4

*Proof.* Define $\tau = \sup_{N_i \in C_N} \frac{\rho}{N_i} g_\rho(N_i, n) = \frac{\rho}{S_N} g_\rho(S_N, n)$, $\Phi_{n,m} = \max\{\Phi_{n_1,m_1}, \Phi_{m_2,n_2}\}$. Since we have $\frac{g_\rho(N,n)}{g_\rho(M,n)} = g_\rho(N, M)$, we could get

$$\left| \frac{\text{GPK}_E[\mathcal{F}, \phi_\rho, X, Y]}{\sqrt{Z}} \right| \leq \left| \frac{\sum_{i,j\in[k]} \left( g_\rho(N_i^{(n_1)}, n) - g_\rho(N_i^{(m_1)}, n) \right) K_{ij} \left( g_\rho(N_j^{(n_2)}, n) - g_\rho(N_j^{(m_2)}, n) \right)}{K_{max} \tau \Phi_{n,m} \sqrt{2n}} \right|$$

$$= \left| \frac{\sum_{i,j\in[k]} \left( g_\rho(N_i^{(n_1)}, n) - g_\rho(N_i^{(m_1)}, n) \right) K_{ij} \left( g_\rho(N_j^{(n_2)}, n) - g_\rho(N_j^{(m_2)}, n) \right)}{K_{max} \frac{\rho}{S_N} g_\rho(S_N, n) \sum_{i\in[k]} \left| g_\rho(N_i^{(n)}, n) - g_\rho(N_i^{(m)}, n) \right| \sqrt{2n}} \right|$$

$$= \left| \frac{S_N \sum_{i,j\in[k]} \left( g_\rho(N_i^{(n_1)}, S_N) - g_\rho(N_i^{(m_1)}, S_N) \right) K_{ij} \left( g_\rho(N_j^{(n_2)}, S_N) - g_\rho(N_j^{(m_2)}, S_N) \right)}{K_{max} \rho \sqrt{2n} \cdot \sum_{i\in[k]} \left| g_\rho(N_i^{(n)}, S_N) - g_\rho(N_i^{(m)}, S_N) \right|} \right|$$

$\square$

## A.5 DETAILS OF REMARK 3.1

$$\text{GPK}_E[\mathcal{F}, \phi_l, X, Y] = \sum_{i\in[k]} \sum_{j\in[k]} \left( g_1(N_i^{(n_1)}, n_1) - g_1(N_i^{(m_1)}, m_1) \right) K_{ij} \left( g_1(N_j^{(n_2)}, n_2) - g_1(N_j^{(m_2)}, m_2) \right)$$

$$= \sum_{i\in[k]} \sum_{j\in[k]} \left( \frac{N_i^{(n_1)}}{n_1} - \frac{N_i^{(m_1)}}{m_1} \right) K_{ij} \left( \frac{N_j^{(n_2)}}{n_2} - \frac{N_j^{(m_2)}}{m_2} \right)$$

$$= \frac{1}{m_1 m_2} \sum_{i=1}^{m_1} \sum_{j=1}^{m_2} k(x_i, x_j) + \frac{1}{n_1 n_2} \sum_{i=1}^{n_1} \sum_{j=1}^{n_2} k(y_i, y_j) - \frac{1}{m_1 n_2} \sum_{i=1}^{m_1} \sum_{j=1}^{n_2} k(x_i, y_j) - \frac{1}{m_2 n_1} \sum_{i=1}^{m_2} \sum_{j=1}^{n_1} k(x_i, y_j)$$

$$= \text{MMD}_l^2[\mathcal{F}, X, Y]$$

## A.6 THE CONVERGENCE BOUNDS OF POWER-MMD WITH $\rho = 1$

When $\rho = 1$, $\tau_{n_1} = \frac{\rho}{N_i} g_\rho(N, n_1) = \frac{1}{N_i} \cdot \frac{N_i}{n_1} = \frac{1}{n_1}$ For simplicity consider the case where $n_1 = n_2 = m_1 = m_2 = n$, from Corollary 3.1. we have

$$\Pr(|\text{GPK}_E[\mathcal{F}, \phi_l, X, Y] - \mathbb{E}[\text{GPK}_E[\mathcal{F}, \phi_l, X, Y]]| \geq a) \leq 2e^{\frac{-na^2}{K_{max}^2 2\Phi^2}}$$

Since $\Phi = \sum_{j\in[k]} |f_\phi(S_{j,n}, S_{j,m})| = \sum_{j\in[k]} \left| \frac{N_j^{(n)}}{n} - \frac{N_j^{(m)}}{m} \right| \leq \sum_{j\in[k]} |\frac{N_j^{(n)}}{n}| + \sum_{j\in[k]} |\frac{N_j^{(m)}}{m}| = 2$

we have

$$\Pr(|\text{GPK}_E[\mathcal{F}, \phi_l, X, Y] - \mathbb{E}[\text{GPK}_E[\mathcal{F}, \phi_l, X, Y]]| \geq a) \leq 2e^{\frac{-na^2}{K_{max}^2 2\Phi^2}} \leq 2e^{\frac{-na^2}{8K_{max}^2}}$$

recall in (Gretton et al., 2012), the deviation bound for linear time estimator $\text{MMD}_l^2[\mathcal{F}, \boldsymbol{p}, \boldsymbol{q}]$ is

$$\Pr(\left| \mathbb{MMD}_l^2[\mathcal{F}, X, Y] - \mathbb{E}[\mathbb{MMD}_l^2[\mathcal{F}, X, Y]] \right| \geq a) \leq 2e^{\frac{-na^2}{8K_{max}^2}}$$

Thus our bound generalize the bound in (Gretton et al., 2012) and provide a tighter version.

## A.7 IS BOUNDS GET TIGHTER WHEN $\rho$ GETTING LARGER?

As we've already known, $\rho = 1$ case is equivalent to $\text{MMD}_l$ in (Gretton et al., 2012), one question rises: Would the performance of cases $\rho > 1$ better than widely used $\rho = 1$ case?

According to Corollary 3.2., since $e^{\frac{-na^2}{8K_{max}^2}} \leq e^{\frac{-na^2}{8\rho^2 K_{max}^2}}$, the convergence bounds for $\rho > 1$ cases seem looser than $\rho = 1$ case, and this may give a negative answer to the question above.

However, the bound above is based on the worst cases where $sup_i(N_i) = n$, such that $\tau_n \leq \frac{\rho}{n}$ and $\Phi \leq 2$. In practice, we are less likely to come across such a phenomena, instead, we may assume the $sup_i(N_i)$ to be far smaller.

Without loss of generality, assume we have $max(\frac{sup_i(N_i^{(n)})}{n}, \frac{sup_i(N_i^{(m)})}{m}) \leq \frac{1}{\alpha}$, where $\alpha \geq 1$, it is easily seen:

$$\tau_n \leq \frac{\rho}{n} \frac{N_i^{\rho-1}}{n^{\rho-1}} \leq \frac{\rho}{\alpha^{\rho-1} n}$$

and

$$\Phi_{n,m} \leq \sum_{i \in [k]} \left(\frac{N_i^{(n)}}{n}\right)^\rho + \sum_{i \in [k]} \left(\frac{N_i^{(m)}}{m}\right)^\rho \leq 2\left(\left(\frac{1}{\alpha}\right)^\rho + \left(1 - \frac{1}{\alpha}\right)^\rho\right)$$

define $\tau := \max\left(\tau_{n_1}, \tau_{n_2}, \tau_{m_1}, \tau_{m_2}\right)$ and $\Phi := \max\left(\Phi_{n_1,m_1}, \Phi_{m_2,n_2}\right)$

We have

$$Z \leq 4n\tau^2\Phi^2 K_{max}^2 \leq 16\frac{\rho^2}{\alpha^{2\rho-2}n}\left(\left(\frac{1}{\alpha}\right)^\rho + \left(1 - \frac{1}{\alpha}\right)^\rho\right)^2 K_{max}^2 = Z_b$$

Plotting the $Z_b$ value with respect to variety value of $\rho$ and $\alpha$ in Fig. 1, we can see that for $\alpha = 1$, the bound will be looser given larger $\rho$. However, for $\alpha$ larger than around 1.25, which means the $sup_i(N_i)$ is slightly smaller than the sample size, the bound will become tighter when $\rho$ is large. This illustrate the benefit of using power-MMD with larger $\rho$ in practice.

We could also get a tighter bound according to Corollary 3.1. Practically, it will be much more beneficial to calculate the $\tau_n = sup_{i \in [k]}\left(\frac{\rho}{N_i^{(n)}}g_\rho(N_i^{(n)}, n)\right)$ and $\Phi_{n,m} = \sum_{i \in [k]}\left|g_\rho(N_i^{(n)}, n) - g_\rho(N_i^{(m)}, m)\right|$ on the fly. That is to say, we do not estimate the convergence bounds before we receive the samples, instead, the calculation of the bounds is carried out together with the calculation of default-plugin-estimators. Remarkably this is still a distribution-free bounds, since we make no assumptions on the probability functions we apply hypothesis test upon.

However, the issue is although $Z$ decreases when $\rho$ increases, $\text{GPK}_E[\mathcal{F}, \phi_\rho, X, Y]$ also decreases when $\rho$ increases. It is not clear how $\rho$ will influence the value of $\left|\frac{\text{GPK}_E[\mathcal{F}, \phi_\rho, X, Y]}{\sqrt{Z}}\right|$.

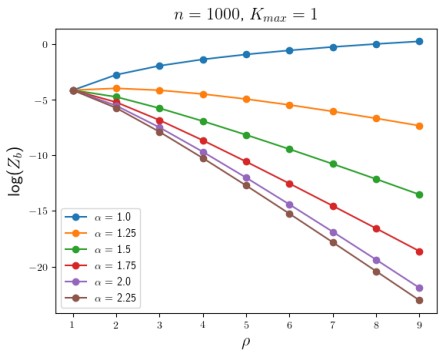

Figure 1: $\log(Z_b)$ with respect to variety of $\rho$ and $\alpha$

