# OpenReview forum: "A generalized probability kernel on discrete distributions and its application in two-sample test"
_ICLR.cc/2021/Conference — Reject_

### Official Review · AnonReviewer2 · 2020-10-25
**new kernels incorporating probability notion are proposed to perform two-sample test; but not yet clearly defined/explained.**

**Rating:** 2
**Confidence:** 4

**Review:**

This paper tries to propose a kernel-based discrepancy measure called generalised probability kernel that can unify MMD and KSD which is an interesting topic of discussion. The paper applies the new discrepancy to perform two-sample tests.
The new kernel proposed, unlike the previous RKHS kernels that only depend on data-points, incorporate the notion of probability. e.g. kernel K_{p,q} depends on density p and q. also a symmetric version on discrete KSD is discussed.
Despite the idea is interesting, there are several flaws which can be reviewed.

Firstly, I think the paper is not clearly presented, with some confusing notation.
--in Definition 1, you defined a kernel, on distributions p and q, that is a k x k matrix; while in definition 2, the notion of K, are on samples and is a scalar output.
it is unclear of how \phi is defined in general; only examples are given later for specific cases so that we got an conjuncture.
--in Definition 5, why is it different from stein operator of KDSD? or it is supposed to say difference operator?

In addition I have several confusions:
1. why is MMD_E^2 an unbiased estimator? what happened to k(x_i, x_i)? it is not clear from the Bernstein polynomial introduced in appendix.
2. in abstract, it claims that the kernels are between distributions instead of samples, but in the main text it is still evaluation at p_i=p(x_i) on samples; I m confused of the difference and novelty claimed.
3. The above concern brings up the question while applying on two sample test.
--When the MMD is used to perform two-sample test, it is assumed that both p and q are unknown. however, to my understanding, we need to know p and q to define k_{prob}; how is this going to be applied to two-sample test?
--for KSD setting, when the symmetric KDSD is introduced, it also seems to require p and q to known for two-sample testing. In the Liu2016 setting, where goodness-of-fit test is proposed with KSD, q is known (up to normalization) while p is unknown with samples; that is a key point why KSD is useful for goodness-of-fit test.
In addition, is there any argument on why the symmetric-KDSD might be better than KDSD Yang et.al 2018?

An additional point is regarding literature review, which is yet throughout  to check; e.g. as
Chwialkowski, et. al  "A kernel test of goodness of fit." proposed independently as Liu et.al for KSD goodness-of-fit test, that might be useful to cite.

In my point of view, ICLR may not be a venue of fit either. More reviews and clarifications may be required, for both kernel construction and application.

---

> ### Author Response · Authors · 2020-11-16
> **we have revised our notation, with more precise definitions.**
>
> Thanks for your reviewing. We have totally remove the part of KSD as we find our proof of theorem 7. in the original version is not correct. In this revision, we mainly focus on polynomial GPK and bring some new results.  From your comment, we noticed that in our first submission, we use K to represents kernel between distributions, kernel between values, and gram matrix. This is confusing, and in this revision we use different terms to denote them.
>
> In response to your comments.
> 1. "in Definition 1, you defined a kernel, on distributions p and q, that is a k x k matrix; while in definition 2, the notion of K, are on samples and is a scalar output."  we define our probability kernel based on a gram matrix, and this gram matrix is produced by a continuous reproducing kernel function as in MMD cases.
>
>
> 2. "it is unclear of how \phi is defined in general; only examples are given later for specific cases so that we got an conjuncture", yes in GPK framework, any element-wise mapping function is valid a function, see Definition 2 in this revision. Our idea is firstly define a framework with little constrains which could generalize a lot of cases. And we then narrow down to specific subsets equipped with some interesting properties. see section 4 of this revision for some examples
>
>
> 3. "why is MMD_E^2 an unbiased estimator?" We find out the plugin-estimator we proposed for GPK framework actually generalize the case of linear time statistics of MMD($MMD_l^2$), and the details of the derivation are in section 6.1 and theorem 2. Also there is a equivalent between the convergence bound of $MMD_l^2$ and our plugin-estimator, see remark3.1
>
>
> 4.  "we need to know p and q to define k_{prob}; how is this going to be applied to two-sample test?" We use plugin-estimators to estimate the GPK[p,q], and once it is a unbiased estimator, no matter what kind of p and q we have, given enough samples, we can always get accuracy estimate of GPK[p,q].(thus we don't need to know p and q beforehand). Furthermore, if GPK[p,q]=0 if and only if p=q, the convergence bound of the estimators could be used to provide acceptance region of null hypothesis p=q, see Corollary 3.3.

---

### Official Review · AnonReviewer3 · 2020-10-26
**The paper has numerous typos, with approximate English and lacks of rigorousness when introducing mathematical concepts; multiple notations are never defined.**

**Rating:** 3
**Confidence:** 4

**Review:**

The paper under review proposes to generalize MMD for discrete random variables whose labels take value in $\mathbb{R}^k$. They propose to estimate these generalized probability kernel distance using empirical estimators. Their properties are studied for two particular examples, namely a kernelized Stein discrenpancy and polynomials versions. Consistency and bias of both estimators are studied and bias corrected.

The paper has numerous typos, with approximate English and lacks of rigorousness when introducing mathematical concepts; multiple notations are never defined. On the theoretical side, the setting on which the contribution relies on is quite strange: in general, labels of discrete variables does not relate to any notion of distance/ordering as in a classical RKHS setting making the relevance of the methodology quite questionable. These point with other technical issues are summarized in following remarks:

## Major points
* As mentioned above, it is quite rare in a discrete setting that the labels lies in $\mathbb{R}$ and satisfies a notion of distance. The authors should better motivate this setting by giving at least on relevant example, either theoretical or practical, where such structure is relevant.
* What does a Stein operator in a discrete setting means? There is a diffenretial operator in Definition 5 that is difficult to generalize and apply in a discrete context.
* The symmetric KDSD introduced in Definition 6 is claimed to be a probability kernel, but the proof that is satisfies Definition 2 is not given.
* The so-called polynomial probability kernel seems to obviously require $l=k$ to satisfy conditions of Definition 3, i.e., that $|\phi(q,p)\| = 0$ implies that $p=q$. It can be called a probability kernel only in such condition.
\end{enumerate}



## Minor points
The paper has numerous typos and imprecisions; a subset of them are listed here.
* p.1: 'underline' should be 'underlying'.
* p.1 when using 'i.e.', always write ',e.i.,'.
* p.1 and onward: there is always a space missing before each parenthesis.
* p.1: Yi & Along (2020) should be a citep and not citet.
* p.1: 'remain futher study' should be for instance 'is left for future work'.
* p.1: KSD is not defined yet.
* p.2: 'the introducting' is `the introduction'.
* p.2 'in representing; is not right.
* p.2 Is $[k]$ the sample space? If yes what is $\{x_1,\dots,x_n\}$? A sample? What is the probability measure $v_i$? Do you mean the probability that $X$ falls in $v_i$?
* p.2 Definition 1: 'Given that distributionS p and q belong ... distributionS with...'. Also 'map' is singular. What is the 'function space' that you refer to? Also where does this definition comes from? Please give proper referencing.
* p.2: Why is there a line break right at the start of 4.1?
* p.2: what is an 'instance of integral probability metric'?
* p.2: last equation $\mu_p$ is not defined, the product opertor $<.,.>_{\mathcal{H}}$ is not defined. $\mathcal{H}$ is not defined.
* p.3: what these 'embedding functions'?
* p.3 the RBH kernel is not defined.
* p.3 second equation: what is $\phi$?
* p.3 Definition 2: the index the sample space should be k, i.e., $y_1,\dots,y_k$ if it refers to the distributions and $n$ for a sample. Here it should be $k$ as it is written distribution.
* p.3: 'examINE', 'members'.
* p.4: the 'brief' proof provided here is only working for discrete variables, while the proof in Gretton et. al deals with continuous variables.
* p.4: what is the 'term above'?
* p.5 'illustrate'
* p.5: there should not be such a thing as an 'art' in science. If you raise that question, then you should formally discuss this topic (choice of optimal $\phi$).
* p.5 Second equation: what are $x_s$ and $x_t$? Notations between this equation and the next are not consistent ($n$ is paired with $x$ and then with $y$ in the next equation).
* p.6: what is this so-called 'same property'?
* p.6: pmfs is never defined.
* p.6 Definition 5: notation $\mathcal{A}$ is never used. $s_p$ is not defined. $\Delta^*$ is not defined. If the latter is a differential operator, what does it means in the context of discrete random variables?
* p.6: what is 'form 5'?
* p.7: what forms 5 and 6?
* p.7: Theorem 5: operator $L$ is not defined. A dot is missing. Are $p$ and$q$ density functions of pmfs?
* p.7: 'preliminary results'.
* p.7: what does 'justing' mean?
* p.8: what is requirement $2$?

---

> ### Author Response · Authors · 2020-11-16
> **we have revised the notation we used, and corrected the typos, hoping this version looks better**
>
> Thanks a lot for your reviewing. We agree our first version is full of typos, and we are sorry for this. In this revision, we have updated the notation section, which provides more precise definitions, and we also checked the typos.
> Since you mentioned citep and citet, we assume ICLR recommend to always use citep instead of citet, is that correct?
> On the theoretical side, yes in the case of categorical distribution, values of discrete variables does not relate to any notion of distance. However, many natural processes will produce discrete distributions where there
> possibly exists a similarity measure in values which imply the similarity in frequencies of occurrence(
> probability values). A good example is the word2vec techniques in NLP tasks. Although the words in a vocabulary is apparently a case of categorical distribution, given enough training data, similarity measure
> between words could be made which implies their similarity in probability values. This is also exactly the case of MMD in discrete setting. And our works is a direct extension of MMD. We introduced this discussion in section 4.2
>
> For your comment related to $l=k$ cases of polynomial GPK, we summarize this result as power-MMD(see section 6). Note that we have also slightly modified our definition of polynomial GPK which generalizes more cases.

---

### Official Review · AnonReviewer4 · 2020-10-27
**Interesting topic, but poorly executed**

**Rating:** 2
**Confidence:** 5

**Review:**

Summary.
The authors describe a family of kernel functions on discrete probability measures. The kernel generalizes existing discrepancies such as the MMD and the KSD. The authors further provide plugin estimates based on empirical frequencies and some arguments for unbiased-ness.

While it is interesting to think about alternative estimators for comparing probability distributions, this paper falls short on the execution. I would recommend a major work-over before considering a submission again. There are many missing points in theory, experiments (there are none), and presentation. See below.

It is unclear to me why we would care about the proposed estimators
* There is no analysis showing that the presented kernels are useful in any way.
* I appreciate that the authors show that existing discrepancies are special cases of the proposed one, but I wonder again what that is useful for?
* This is in particular as the authors do not provide any sort of asymptotic analysis of the presented estimators. How can we use them for two-sample testing without that? Answering this question is one of the major parts of the kernel two-sample testing literature.

Theory.
* The presented theory consists of elementary manipulations that mostly follow existing literature, so there is very little actual innovation. For example of of page 5.

Experimental evaluation
* There is *no* experimental evaluation of the proposed estimators.
* It would have been interesting to compare the variances as a function of dataset characteristics.

Presentation
* There many grammar glitches, spelling mistages, missing articles, etc, to a point that it is hard to follow.
* There is no overview of the series of arguments in the later part of the paper.
* There is a lot of re-cited derivations from existing papers.

---

> ### Author Response · Authors · 2020-11-16
> **we have new results with convergence bound, and based on the GPK framework, we propose a new statistics for two-sample test with better performance than linear time MMD**
>
> Thank you for your reviewing. In this revision, we present the usage of our GPK framework in proposing new statistics for two-sample test, which we call power-MMD. power-MMD is a direct extension of MMD in the framework of GPK. We provide unbiased estimator and convergence bounds of it.
>
> As to KSD, since we found that our proof in theorem 7.(related with KSD) in our first version of paper is not correct, we do not have any new result related with KSD anymore. Thus we totally remove the discussion of KSD in this first revision.
>
> We also revise the presentation, including the typos and math notations, hoping this version would look better.

---

### Official Review · AnonReviewer1 · 2020-10-28
**Seems to be an incomplete submission with missing details**

**Rating:** 1
**Confidence:** 5

**Review:**

The works proposes a generalization of MMD-squared distance. However, the submission seems to be an incomplete one.

Major comments:
1. Definition 2 seems to be the key definition in the work. However, there are multiple issues:
       a. It is not clear why it is called a kernel? Should not it be called distance? After all, it generalizes MMD-squared!
       b. "$K$" seems to be mixed up with "$k$". "$K$" seems to be the gram matrix and not the kernel.
       c. $k(y_i, y_j)$ is from $Y \times Y\rightarrow R$, and not from $R^n \times R^n \rightarrow R$.
       d. why should "\phi" belong to RKHS of k? Recall that RKHS of k will contain functions from R^n\rightarrow R.
2. I agree with the write-up which states  that results in section 4.2 are trivial.
3. section 4.3.1 are known results and need to be skipped.
4. Proof of theorem 5 seems to be completely missing. How is sup over f removed?
5. In Definition 6, what is $f$? Is a sup over f missing? Because of this and the previous issue, section 5.1 seems very incomplete.
6. Simulation section seems to be completely missing.
7. Connection with Bernstien polynomials highlighted in intro etc. seems to be missing.

---

> ### Author Response · Authors · 2020-11-16
> **we have revised the submission, with some valuable new results**
>
> Thank you for reviewing our submission. We admit that our first version of paper has a lot of problem, hoping this revision would be better. In response to the comments:
> 1. a. we call it a kernel because it is a more general definition, which include the cases where GPK[p,q] increase when p and q become similar. Since the mapping function $\phi$ allows a great number of possibilities, we do not know if every case satisfies the requirement of distance measure.
> 1. b. d. we have modified the definition, and we think this time it will be clear
> 1. c. we assume the values of the discrete distribution Y are in d dimensional space R^d
> 2.3.4.5 we modified our theory and have new results related with polynomial GPK
> 7. Bernstein polynomial is used to search for unbiased estimators, see theorem 2. in this revision.

---

### Author Response · Authors · 2020-11-16
**description of first revision**

As we found that our proof in theorem 7.(related with KSD) in our first version of paper is not correct, we do not have any new result related with KSD anymore. Thus we totally remove the discussion of KSD in this first revision. We then focus on the discussion of polynomial GPK. We generalize the result of unbiased estimator in our first version into a more general theorem(theorem 2), and provide convergence bound of the unbiased estimator we discovered(theorem 3). We apply the result to a special case of polynomial GPK, which we call it power-MMD. We illustrate that power-MMD could also be used for two-sample test.
In our first submission, we use K to represents kernel between distributions, kernel between values, and gram matrix. This is confusing, and in this revision we use different terms to denote them. We also update the Notation section to have a more accuracy description of the terms we introduced.

---

### Decision · Program_Chairs · 2021-01-07
**Final Decision**

**Decision:**

Reject

**Comment:**

The focus of the submission is to define divergences on discrete probability measures. Particularly, the authors propose a common generalization of the well-known concept of maximum mean discrepancy and kernel Stein discrepancy.

As summarized by the reviewers the submission is in a rather preliminary form:
1)The work lacks motivation.
2)Literature review (there are 4 references in total) and numerical illustrations are missing.
3)The submission lacks proper mathematical formulation/rigor.
I highly recommend the authors to not submit similar draft manuscripts in the future.